

# Heavy grazing reduces soil bacterial diversity by increasing soil pH in a semi-arid steppe

Xiaonan Wang, Chengyang Zhou, Shining Zuo, Yixin Ji, Wenxin Liu and Ding Huang

College of Grassland Science and Technology, China Agricultural University, Beijing, China

## ABSTRACT

**Background**. In a context of long-term highly intensive grazing in grassland ecosystems, a better understanding of how quickly belowground biodiversity responds to grazing is required, especially for soil microbial diversity.

**Methods**. In this study, we conducted a grazing experiment which included the CK (no grazing with a fenced enclosure undisturbed by livestock), light and heavy grazing treatments in a desert steppe in Inner Mongolia, China. Microbial diversity and soil chemical properties (*i.e.*, pH value, organic carbon, inorganic nitrogen (IN, $NH_4^+$-N and $NO_3^-$-N), total carbon, nitrogen, phosphorus, and available phosphorus content) both in rhizosphere and non-rhizosphere soils were analyzed to explore the responses of microbial diversity to grazing intensity and the underlying mechanisms.

**Results**. The results showed that heavy grazing only deceased bacterial diversity in the non-rhizosphere soil, but had no any significant effects on fungal diversity regardless of rhizosphere or non-rhizosphere soils. Bacterial diversity in the rhizosphere soil was higher than that of non-rhizosphere soil only in the heavy grazing treatment. Also, heavy grazing significantly increased soil pH value but deceased NH4+-N and available phosphorus in the non-rhizosphere soil. Spearman correlation analysis showed that soil pH value was significantly negatively correlated with the bacterial diversity in the non-rhizosphere soil. Combined, our results suggest that heavy grazing decreased soil bacterial diversity in the non-rhizosphere soil by increasing soil pH value, which may be due to the accumulation of dung and urine from livestock. Our results highlight that soil pH value may be the main factor driving soil microbial diversity in grazing ecosystems, and these results can provide scientific basis for grassland management and ecological restoration in arid and semi-arid area.

## INTRODUCTION

Grasslands are the important terrestrial ecosystems which support multiple ecosystem function and services (*e.g.*, plant productivity and livestock production) (*Nan, 2005*; *Pan et al., 2018*; *Fan et al., 2021a*); however, frequent or continuous intensification or de-intensification of management (*e.g.*, high or low grazing intensities) in grasslands may deeply affect soil system and its microbial component (*Attard et al., 2008*). To date, grasslands have been degraded to various extents due to highly intensive management

Corresponding author
Ding Huang, huangding@263.net

(*Eldridge & Delgado-Baquerizo, 2017*; *Wang et al., 2020*), such as grazing and haying, which may deteriorate the soil physical, chemical and biological properties (*e.g.*, change in bulk density, organic carbon, nitrogen, microbial diversity in soil systems) (*Northup, Brown & Holt, 1999*; *Steffens et al., 2008*; *Chen et al., 2016a*; *Fan et al., 2021b*). Concerns over high intensive grazing disturbance is also reasonable, as this also may cause significant degradation due to loss of sensitive species, or removal of nutrients (*Zhang et al., 2022*). Conversely, low intensive grazing may have positive effects on soil system and its microbial component (*Liu et al., 2015*; *Wang et al., 2022b*), due to the return of dung and urine from livestock. Appropriate management of these ecosystems thus requires understanding how low or high intensive grazing regimes will affect the soil system and its microbial component.

Soil microorganisms (*e.g.*, bacteria and fungi) are the vital component of biodiversity and play important roles in the grassland ecosystem, such as litter decomposition, soil nutrient supply, the elements' biogeochemical cycle, which regulate the plant nutrient availability (*Bardgett, 2005*; *Yao, Bowman & Shi, 2011*; *Zhang et al., 2016*; *Zhao et al., 2020*; *Wang et al., 2022a*). The loss of microbial diversity leads to the decline in ecosystem functions, which negatively impacts on primary production, nutrient cycling and climate regulation (*Delgado-Baquerizo et al., 2016*). Microbial diversity have been shown to be strongly influenced by soil physical and chemical properties (*e.g.*, soil pH value, organic carbon, moisture, and *etc.*; *Chen et al., 2017*; *Ren et al., 2018*; *Wang et al., 2020*; *Wang et al., 2021*), vegetation (*Wu et al., 2022*), climate (*Singh et al., 2010*; *Sheik et al., 2011*) or other mechanisms. In livestock grazing systems, microbial diversity may be driven by different underlying mechanisms due to the complex effects of livestock grazing on grassland systems (*i.e.*, feeding, dung and urine return, and trampling; *Liu et al., 2015*).First, feeding effect of livestock grazing may reduce the above-ground biomass and affect below-ground biased allocation, as well as change in root exudates, then may affect soil microbial community (*Yang et al., 2013*; *Liu et al., 2015*; *Chung & Rudgers, 2016*; *Fan et al., 2021a*). Second, the return of dung and urine from livestock may alter soil microbial community by changing soil nutrient availability (*Kohler et al., 2005*; *Bai et al., 2012*; *Jing et al., 2023*). Third, trampling effect of livestock grazing may compact the soil and change soil bulk density, soil water potential, aeration, and redox conditions, thus alter soil microbial community (*Kauffman, Thorpe & Brookshire, 2004*; *Liu et al., 2015*; *Wang & Wesche, 2016*; *Fan et al., 2021a*; *Jing et al., 2023*).

Previous studies have reported that grazing may affect soil microbial diversity (*Zhang et al., 2016*; *Wang et al., 2022a*; *Jing et al., 2023*), however, findings have been mixed. *Wu et al. (2022)* conducted a regional-scale grazing experiment on the Mongolian Plateau (covered three typical vegetation types) and showed that grazing increased the diversity of both bacterial and fungal compared with enclosure. Conversely, largely previous studies have found that grazing had negative (*Fan et al., 2021a*; *Wang et al., 2022b*) or no (*Fan et al., 2021b*; *Jing et al., 2023*) effects on soil bacterial diversity. For example, *Wang et al. (2022b)* conducted a grazing experiment with five grazing intensities in Inner Mongolian desert steppe and found that highly intensive grazing (*i.e.*, heavy and over-grazing) significantly reduced the diversity of both bacteria and fungi, due to the change in soil organic content

and pH value. As these studies indicate, the effects of grazing on the soil microbial diversity may change depending on grazing intensity, grassland types or others (*Hu et al., 2019*), and these factors should be considered when managing these ecosystems.

Desert steppe accounts for 39% of the total grassland area in Inner Mongolia (*Han et al., 2008*; *Zhang et al., 2018a*; *Zhang et al., 2023*), where plant productivity are low due to low precipitation and temperature and is very sensitive to grazing pressure (*Olofsson, 2006*; *Fan et al., 2021a*). From the perspective of soil microorganisms, it is of great significance to reveal the disturbance mechanism of grazing for the health and sustainable development of grassland ecosystem. It is unclear, however, whether low or high intensive grazing intensity will affect the diversity of soil microbes in these grasslands. To test the effect of varying grazing intensity on the diversity of soil microbes, we conducted an experiment in a desert steppe of northern China. We thus tested whether light or heavy grazing would affect the diversity of soil microbes, and which soil factors drive the changes of microbial diversity.

## MATERIALS & METHODS

### Study site and experimental design
Our experiment was performed in the Xilamuren desert steppe (41°15′10″N, 111°13′26″E), Inner Mongolia, China. The mean annual precipitation and temperature are 280 mm and 2.5 °C, respectively. A dominant plant is *Stipa krylovii*, while *S. breviflora*, *Cleistogenes songorica* and *Artemisia frigida* are common species in this area. The soil type at this site is loamy sand texture.

A randomized complete block design was utilized in this experiment. Three blocks were established and each of them contained three treatments, with the total of 9 plots in the experimental area. The three treatments were no grazing (with a fenced enclosure undisturbed by livestock; CK), light grazing (with a stocking rate of 0.63 sheep/ha/year (3 horses); LG) and heavy grazing (with a stocking rate of 1.05 sheep/ha/year (5 horses); HG). The grazing platform has been implemented in each growing season (*i.e.,* from June to September) since 2012. Adult Mongolian horses were selected to graze. The detailed information on study site and experimental design can be found in our previous study (*Wang et al., 2022c*).

### Sampling and measurement
In August 2021, we assessed species composition, and the height, density, individual biomass and relative density of *S. krylovii* population based on ten 1 m × 1 m quadrats per plot. Then, we randomly selected 30 clumps of *S. krylovii* (30 cm soil layer each plant individual) in each plot, removed the loose soil on the surface of roots, and obtained the rhizosphere soil (RS) by the shaking method. Every ten samples were mixed into one composite sample. At the same time, ten non-rhizosphere soils (NRS) of equal depth were excavated along the Z-shape and mixed into one composite sample in each plot. In addition, we collected other soil cores for soil bulk density. The composite samples were passed through a two mm sieve and divided into two parts. One part was naturally dried at room temperature for the determination of soil chemical properties except for soil inorganic nitrogen content with fresh soil, and the other part was placed at −80 °C

for the determination of microbial diversity. Data on inorganic nitrogen ($NH_4^+$-N and $NO_3^-$-N), soil pH, total carbon (TC), total nitrogen (TN), soil organic carbon (SOC), total phosphorus (TP), available phosphorus (AP) and soil bulk density were collected as previously described in *Zhang et al. (2020a)*.

## Determination of soil microbial diversity

After the soil sample was frozen for two days, the DNA of microbial was extracted from soil using the PowerSoil DNA Isolation Kit (MO BIO laboratories, Carlsbad, CA, USA). The DNA extract was determined on 1% agarose gel, and the concentration and purity of DNA were examined using a NanoDrop 2000 UV-Vis spectrophotometer (Thermo Fisher Scientific, Waltham, MA, USA). The V4 region of the 16S gene was sequenced by PCR: 515 F (GTGCCAGCMGCCGCGGTAA) and 806 R (GGACTACHVGGGTWTCTAAT), while PCR: ITS5-1737F (CTTGGTCAT TTAGAGGAAGTAA) and ITS2-2043R (GCTGCGTTCTTCATCGATGC) was sequenced for the ITS1 region gene. The PCR product was tested in a 2% agarose gel. The construction of library was established in a TruSeq® DNA PCR-Free Sample Preparation Kit. The sequencing work was completed by Beijing NuoHeZhiYuan Technology Co. Ltd. The 16S rRNA and ITS rRNA sequences are available at NCBI Sequence Read Archive (SRA) database (Accession Number: PRJNA1014180 and PRJNA1014553).

## Statistical analysis

To test the interaction effects between grazing intensities (*i.e.,* CK, LG and HG) and soil positions (both in rhizosphere and non-rhizosphere soils) treatments on the microbial diversity and soil chemical properties, we used a two-way ANOVA with grazing treatments as the main-plot effect, and soil position treatments as the sub-plot effect. Microbial diversity was calculated based on OTUs number, including Richness, Shannon-Wiener index and Chao1 index, calculated by the following formula:

Richness $= S_{obs}$
Shannon-Wiener index: $H_{shannon} = -\sum_{i=1}^{S_{obs}} \frac{n_i}{N} \ln \frac{n_i}{N}$
Chao1 index: $S_{chao1} = S_{obs} + n_1(n_1 - 1)/2(n_2 + 1)$.

Where: $S_{obs}$ is the OTUs number actually observed; $S_{chao1}$ is the estimated OTUs number; $n_1$ represents the number of OTUs containing only one sequence; $n_2$ represents the number of OTUs containing only two sequences; $n_i$ is the number of sequences contained in the i th OTUs; N represents all the sequence numbers. Besides, spearman correlation analysis was used to assess the relationships between the microbial diversity (*i.e.,* both bacterial and fungal) and soil chemical properties. Data analysis was conducted using SPSS 19.0 (IBM Corp., Armonk, NY, USA), and all figures were prepared using Origin 2021 (OriginLab Corporation, Northampton, MA, USA).

**Table 1  Effects of grazing treatment (G), soil position (SP) and their interaction on soil microbial diversity and chemical properties in a desert steppe ecosystem in China.**

| Resource | | Grazing (G) | | Soil position (SP) | | G * SP | |
|---|---|---|---|---|---|---|---|
| | | *F* | *P*-value | *F* | *P*-value | *F* | *P*-value |
| Bacteria | Richness | 26.118 | <0.001 | 26.974 | <0.001 | 21.329 | <0.001 |
| | Shannon | 17.917 | <0.001 | 10.964 | 0.002 | 12.468 | <0.001 |
| | Chao1 | 13.907 | <0.001 | 12.038 | 0.002 | 10.936 | <0.001 |
| Fungi | Richness | 1.268 | 0.296 | 2.884 | 0.1 | 3.656 | 0.038 |
| | Shannon | 0.297 | 0.745 | 1.933 | 0.175 | 1.937 | 0.162 |
| | Chao1 | 2.124 | 0.137 | 0.573 | 0.455 | 3.084 | 0.061 |
| Chemical properties | pH | 20.648 | <0.001 | 392.225 | <0.001 | 17.944 | <0.001 |
| | SOC (g kg$^{-1}$) | 0.165 | 0.85 | 5.029 | 0.045 | 0.055 | 0.947 |
| | Total P (g kg$^{-1}$) | 12.194 | 0.001 | 2.522 | 0.138 | 3.149 | 0.08 |
| | Available P (mg kg$^{-1}$) | 13.508 | 0.001 | 41.296 | <0.001 | 1.886 | 0.194 |
| | $NH_4^+$-N (mg kg$^{-1}$) | 66.204 | <0.001 | 569.775 | <0.001 | 42.76 | <0.001 |
| | $NO_3^-$-N (mg kg$^{-1}$) | 62.406 | <0.001 | 313.895 | <0.001 | 41.127 | <0.001 |
| | Inorganic N (mg kg$^{-1}$) | 64.573 | <0.001 | 863.851 | <0.001 | 66.03 | <0.001 |
| | Total C (g kg$^{-1}$) | 1.803 | 0.207 | 62.105 | <0.001 | 0.165 | 0.85 |
| | Total N (g kg$^{-1}$) | 2.382 | 0.135 | 33.891 | <0.001 | 2.016 | 0.176 |

# RESULTS

## The interaction effects between grazing intensities and soil positions on both bacterial and fungal community diversity

Heavy grazing decreased diversity (*i.e.*, richness, Shannon and Chao1) of bacterial community in the non-rhizosphere soil ($F = 26.915$, $P = 0.001$, $F = 10.96$, $P = 0.01$, $F = 8.625$, $P = 0.017$, for richness, Shannon and Chao1, respectively; Figs. 1A, 1C and 1E), however, there had no significant differences in diversity (*i.e.*, richness, Shannon and Chao1) of bacterial community among grazing treatments in the rhizosphere soil ($F = 0.514$, $P = 0.604$, $F = 1.505$, $P = 0.242$, $F = 0.336$, $P = 0.718$, for richness, Shannon and Chao1, respectively; Figs. 1A, 1C and 1E). Overall, bacterial diversity (*i.e.*, richness and Chao1) in the rhizosphere soil was higher than that of non-rhizosphere soil only in the heavy grazing treatment ($P < 0.001$ and 0.042 for richness and Chao1, respectively; Figs. 1A and 1E). There was significant variation in bacterial diversity (*i.e.*, richness, Shannon and Chao1) among grazing intensity treatments, as well as soil position treatments, and a significant interaction between grazing intensity and soil position treatments (Table 1).

For fungal community, however, there had no significant differences in fungal diversity (*i.e.*, richness, Shannon and Chao1) among grazing treatments both in the rhizosphere and non-rhizosphere soils (for the rhizosphere soil, $F = 0.814$, $P = 0.455$, $F = 0.769$, $P = 0.474$, $F = 0.774$, $P = 0.472$, for richness, Shannon and Chao1, respectively, for the non-rhizosphere soil, $F = 3.163$, $P = 0.115$, $F = 1.199$, $P = 0.365$, $F = 3.93$, $P = 0.081$, for richness, Shannon and Chao1, respectively; Figs. 1B, 1D and 1F). Likewise, there had no significant differences in fungal diversity (*i.e.*, richness, Shannon and Chao1) between rhizosphere and non-rhizosphere soil in all grazing intensity treatments (all $P > 0.05$; Figs.

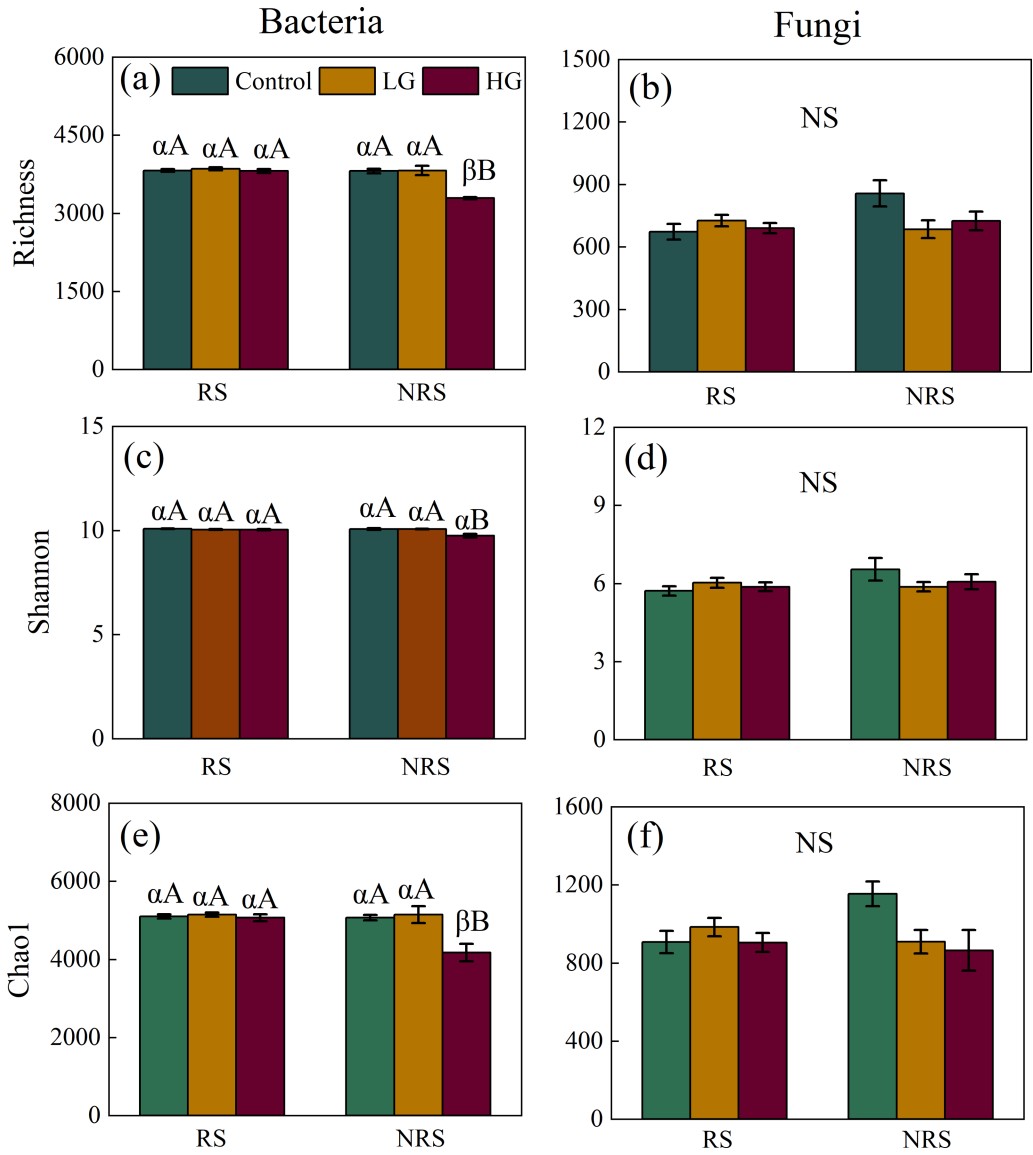

**Figure 1** **Interaction effect between grazing intensity and soil position treatments on diversity of both bacteria and fungi in a desert steppe ecosystem in China.** (A) Richness of bacteria, (B) richness of fungi, (C) Shannon–Wiener index of bacteria, (D) Shannon–Wiener index of fungi, (E) Chao1 of bacteria, and (F) Chao1 of fungi. Error bars represent s.e.m. Significance levels: non-significance (NS) for $P > 0.05$. Different uppercase letters indicate significant differences ($P < 0.05$) among different grazing treatments in same soil position (*i.e.,* rhizosphere or non-rhizosphere soil), and different Greek letters indicate significant differences ($P < 0.05$) between rhizosphere and non-rhizosphere soil in same grazing treatment (*i.e.,* CK, LG or HG treatment). CK, fenced with no grazing, LG, light grazing, HG: heavy grazing.

1B, 1D and 1F). There had no significant interaction in fungal diversity (*i.e.,* Shannon and Chao1) between grazing intensity and soil position treatments, except for richness of fungal community (Table 1).

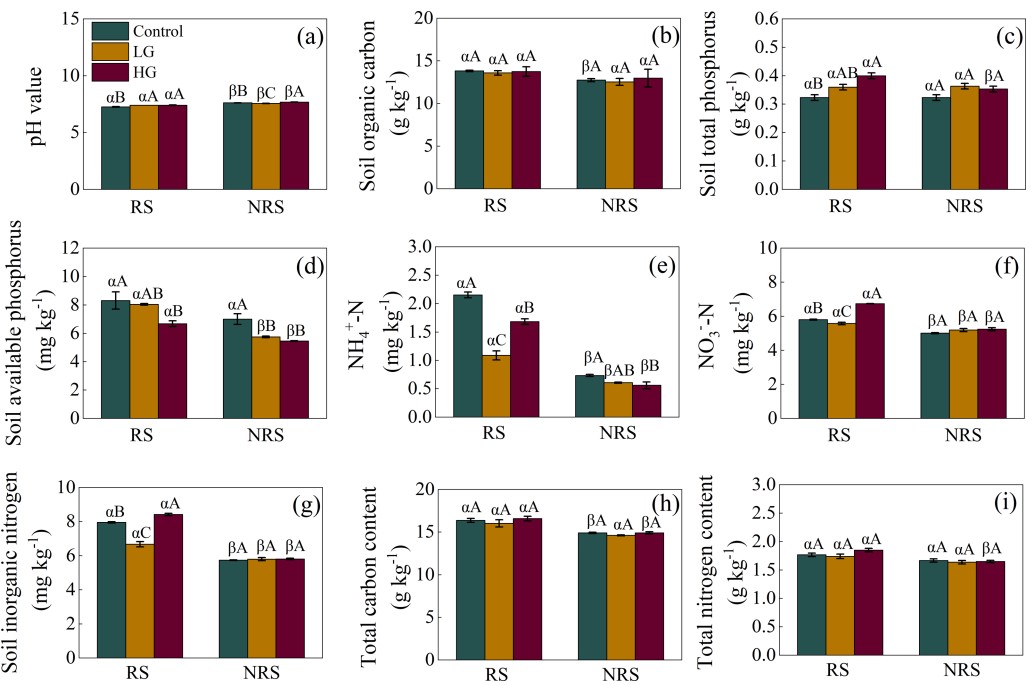

**Figure 2 Interaction effect between grazing intensity and soil position treatments on soil chemical properties in a desert steppe ecosystem in China.** (A) pH, (B) soil organic carbon, (C) total phosphorus, (D) available phosphorus, (E) ammonium nitrogen ($NH_4^+$-N), (F) nitrate nitrogen ($NO_3^-$-N), (G) inorganic nitrogen, (H) total carbon, (I) total nitrogen. Error bars represent s.e.m. Significance levels: non-significance (NS) for $P > 0.05$. Different uppercase letters indicate significant differences ($P < 0.05$) among different grazing treatments in same soil position (*i.e.*, rhizosphere or non-rhizosphere soil), and different Greek letters indicate significant differences ($P < 0.05$) between rhizosphere and non-rhizosphere soil in same grazing treatment (*i.e.*, CK, LG or HG treatment). CK, fenced with no grazing, LG, light grazing, HG, heavy grazing.

## The interaction effects between grazing intensities and soil positions on soil chemical properties

For the rhizosphere soil, light grazing increased soil pH value but decreased $NH_4^+$-N, $NO_3^-$-N and inorganic N content ($F = 14.849$, $P = 0.005$, $F = 74.575$, $P < 0.001$, $F = 184.481$, $P < 0.001$, $F = 91.036$, $P < 0.001$, for soil pH value, $NH_4^+$-N, $NO_3^-$-N and inorganic N content, respectively; Figs. 2A, 2E, 2F and 2G), while heavy grazing increased soil pH value, total P, $NO_3^-$-N, and inorganic N content ($F = 14.849$, $P = 0.005$, $F = 11.679$, $P = 0.001$, $F = 184.481$, $P < 0.001$, $F = 91.036$, $P < 0.001$, respectively; Figs. 2A, 2C, 2F and 2G), but decreased available P and $NH_4^+$-N ($F = 5.575$, $P = 0.043$, $F = 74.575$, $P < 0.001$, respectively; Figs. 2D and 2E). For the non-rhizosphere soil, however, light grazing decreased soil pH value and available P ($F = 42.304$, $P < 0.001$, $F = 13.561$, $P = 0.006$, respectively; Figs. 2A and 2D), while heavy grazing increased soil pH value but decreased available P and $NH_4^+$-N content ($F = 42.304$, $P < 0.001$, $F = 13.561$, $P = 0.006$, $F = 5.184$, $P = 0.049$, respectively; Figs. 2A, 2D and 2E).

Besides, soil pH value in the non-rhizosphere soil was higher than that of rhizosphere soil in all grazing treatments ($P = 0.007$, $P = 0.003$, $P < 0.001$, for control, LG and HG

treatment, respectively; Fig. 2A). Conversely, soil $NH_4^+$-N, $NO_3^-$-N, inorganic N content in the non-rhizosphere soil was lower than that of rhizosphere soil in all grazing treatments (all $P < 0.001$ in the control, $P = 0.023$, $0.027$ and $0.011$ in the light grazing treatment, all $P < 0.001$ in the heavy grazing treatment for $NH_4^+$-N, $NO_3^-$-N, inorganic N content, respectively; Figs. 2E, 2F and 2G). Besides, soil organic carbon in the non-rhizosphere soil was lower than that of rhizosphere soil only in the control treatment ($P = 0.017$; Fig. 2B), while both soil total P and total N content in the non-rhizosphere soil was lower than that of rhizosphere soil only in the heavy grazing treatment ($P = 0.049$ and $0.004$, respectively; Figs. 2C and 2I). Further, soil available P content in the non-rhizosphere soil was lower than that of rhizosphere soil both in the light and heavy grazing treatment ($P < 0.001$, $P = 0.003$, respectively; Fig. 2D), while total C content in the non-rhizosphere soil was lower than that of rhizosphere soil both in the control and heavy grazing treatment ($P = 0.018$ and $0.012$, respectively; Fig. 2H). A significant interaction between grazing intensities and soil position treatments in soil pH value, $NH_4^+$-N, $NO_3^-$-N, inorganic N content; however, there had no significant interaction in soil organic carbon, available P, total C, N and P content (Table 1).

## Relationships between diversity of microbial community and soil chemical properties

In the non-rhizosphere soil, the diversity of bacterial community (*i.e.,* richness, Shannon and Chao1) was significantly negatively correlated with the soil pH value (all $P < 0.05$, $r = -0.80$, $-0.67$ and $-0.78$, respectively; Fig. 3). For fungal community, however, the Chao1 was only significantly positively correlated with available P content in the non-rhizosphere soil ($r = 0.76$, $P > 0.05$; Fig. 3). Also, soil available P content was significantly positively correlated with $NH_4^+$-N content but negatively correlated with $NO_3^-$-N content (all $P > 0.05$; $r = 0.82$ and $-0.76$, respectively; Fig. 3), while soil $NH_4^+$-N content was significantly negatively correlated with $NO_3^-$-N content ($r = -0.87$, $P > 0.05$; Fig. 3). In the rhizosphere soil, soil $NH_4^+$-N content was significantly negatively correlated with the richness of bacteria and fungi, soil pH value (all $P < 0.05$, $r = -0.80$, $-0.70$ and $-0.80$, respectively; Fig. S1). Soil total P in the rhizosphere soil was significantly positively correlated with soil pH but negatively correlated with soil available P (all $P < 0.05$, $r = 0.67$ and $-0.73$, respectively; Fig. S1). Soil total N in the rhizosphere soil was significantly negatively correlated with soil available P but positively correlated with total C (all $P < 0.05$, $r = -0.74$ and $0.70$, respectively; Fig. S1).

## DISCUSSION

The results of our grazing experiment demonstrated that decreased bacterial diversity in the non-rhizosphere soil from heavy grazing owns to the increase of soil pH value in the Inner Mongolian steppe. These results highlight that soil properties may play a role in the regulation of soil microbial diversity, and our results also suggest that the main drivers should be given more attention, especially in complex grazing systems, where feeding, dung and urine return, and trampling effects of livestock grazing may have direct or indirect potential effects on different soil factors. In this complex grazing systems, the feeding of

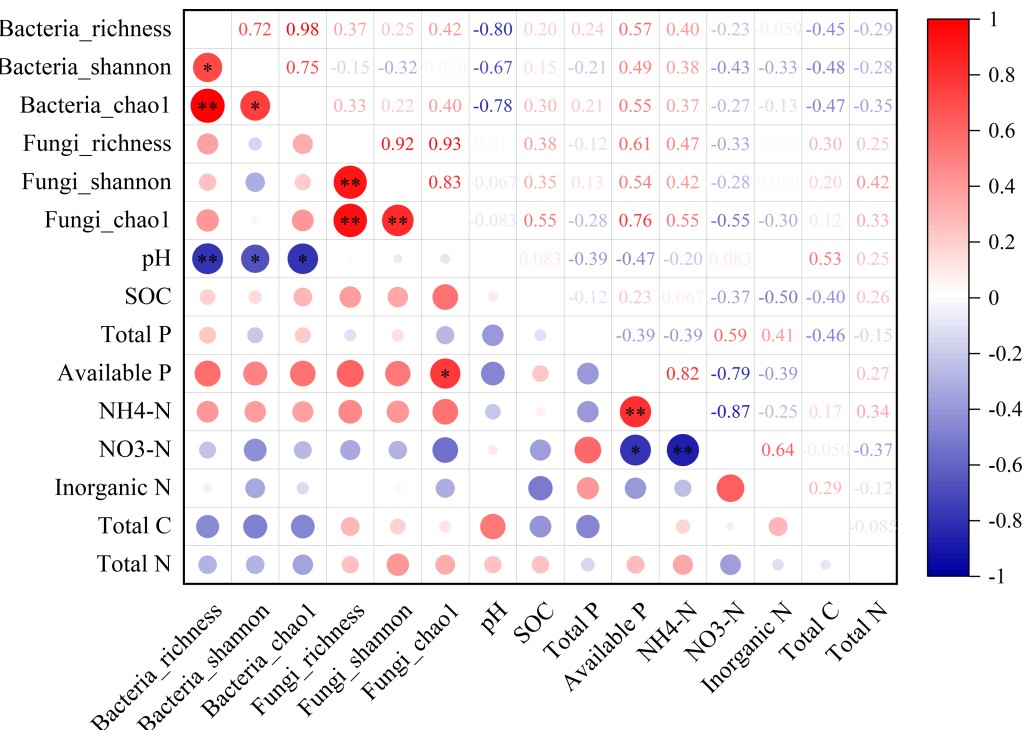

**Figure 3 The relationship between microbial diversity (both bacteria and fungi) and soil chemical properties in the non-rhizosphere soil in a desert steppe ecosystem in China.** Red and blue denote positive and negative correlations, respectively (* $P < 0.05$; ** $P < 0.01$).

livestock had negative effect on population dynamics characteristic of *S. krylovii* (*i.e.,* the reduction in height and individual biomass); however, there was no significant effect of livestock trampling that soil bulk density was not significant among treatments (Figs. S2 and S3). We speculated that the increase in pH value under heavy grazing plot may largely be driven by the effect of dung and urine return, then negatively affects soil bacterial diversity.

Our results provided evidence that heavy grazing significantly reduced the diversity of bacteria, which is in line with another grazing experiment in the desert steppe of Inner Mongolia (*Wang et al., 2022b*). We speculated that this may be due to the differences in the competitive abilities of different microbial phyla (*Zhang et al., 2020a*; *Wang et al., 2022b*). Conversely, *Fan et al. (2021b)* found that heavy grazing had no significant effects on bacterial diversity in a desert steppe, because grazing did not change soil properties (*e.g.,* soil pH value, soil organic carbon content), which largely affected the change of bacterial diversity. In our study, heavy grazing reduced the diversity of bacteria, available P and $NH_4^+$-N content, and increased soil pH value in the non-rhizosphere soil, while bacterial diversity was significantly related to soil pH value. Similarly, *Qu et al. (2016)* provided some evidences that soil pH value significantly increased under heavy grazing treatment in the meadow steppe. We speculated that increased non-rhizosphere soil pH value in the heavy grazing treatment may be due to the return of dung and urine from livestock grazing

(horse in our study). High soil pH value was considered as a special environment from heavy grazing-induced, which resulted in few adapted taxa (*Fierer, Bradford & Jackson, 2007*; *Qu et al., 2016*). However, previous studies have found that soil nitrogen was the most pronounced factor shaping soil bacterial community (*Wang et al., 2021*; *Zhang et al., 2020b*), and may affect the growth and survival of the bacteria through offering the substrate (*Wang et al., 2019*; *Wang et al., 2021*). The differences between our study and other experiments also may be due to the differences in grazing intensity, grassland types or livestock (*e.g.*, sheep, cow or horse) (*Hu et al., 2019*). Previous studies have revealed that microbial may be driven by soil physical and chemical properties (*e.g.*, soil pH value, organic carbon; *Chen et al., 2017*; *Ren et al., 2018*; *Wang et al., 2020*; *Wang et al., 2021*). The return of dung and urine from livestock may alter soil microbial community by changing soil nutrient availability or other properties (*e.g.*, soil pH value; *Kohler et al., 2005*; *Bai et al., 2012*; *Jing et al., 2023*).

Contrary to bacterial community showing that soil fungal diversity did not significantly change among grazing intensity treatments (control to heavy grazing), regardless of rhizosphere or non-rhizosphere soil, which is in line with a grazing and enclosure experiment on the Mongolian Plateau that soil fungi diversity did not differ between grazed and ungrazed (*Wu et al., 2022*). This may be due to the difference in sensitivity to external disturbances (*e.g.*, grazing), while bacterial community are more sensitive to grazing than fungal community (*Cheng et al., 2016*; *Zhang et al., 2018b*; *Wu et al., 2022*).

Another interesting thing is that soil bacterial diversity in the non-rhizosphere soil in the heavy grazing treatment was lower than that in the rhizosphere soil, which is in line with some previous studies (*Edwards et al., 2015*; *Dawson et al., 2017*; *Novello et al., 2017*; *Schöps et al., 2018*). Plant rhizosphere is the active region controlling nutrients transformation (*Zhang et al., 2010*; *Ahkami et al., 2017*), and differs from the non-rhizosphere soil (*Chen et al., 2016b*; *Liu et al., 2023*). The reason may be that the microbial communities in the rhizosphere soil are more differentiated and contain a high number of specialists compared to the non-rhizosphere soil (*Mendes et al., 2014*; *Schöps et al., 2018*). Plants provide a pulse of readily available carbon substrates into soils when we dug them (*Bardgett et al., 2005*; *Schöps et al., 2018*). Different composition of these compounds may promote the diversity of microbial community and lead to strong niche differentiation (*Hinsinger et al., 2005*; *Schöps et al., 2018*).

## CONCLUSIONS

Heavy grazing reduced the diversity of bacterial community in the non-rhizosphere soils, driven by increased soil pH value, though having no effect on fungal diversity both in the rhizosphere and non-rhizosphere soil. Our results highlight that the negative effect of highly intensive grazing on bacterial community, and the bacterial community may be more sensitive to grazing disturbance than fungal community. Also, soil bacterial diversity in the non-rhizosphere soil from heavy grazing was lower than that in the rhizosphere soil, and this may give us some insight to uncover the differences among different soil position and its underlying mechanisms. Combined, our results suggest that decreased soil bacterial

diversity in the non-rhizosphere soil from heavy grazing may own to the increase of soil pH value, which may largely be due to the accumulation of dung and urine from livestock. Since the experiment lasted only eight years, long-term effects will be studied in the future.

## ACKNOWLEDGEMENTS

We thank many students for continuously collecting data in the field and for the laboratory analysis. We thank the Key Laboratory of Grassland Resources of the Ministry of Education (Inner Mongolia Agricultural University) for technical support.

### Funding

This work was supported by the earmarked fund for CARS (CARS-34) and the Climate-Smart Grassland Ecosystem Management Project (10006-P166853-2021-PIR-WB-China). There was no additional external funding received for this study. The funders had no role in study design, data collection and analysis, decision to publish, or preparation of the manuscript.

### Grant Disclosures

The following grant information was disclosed by the authors:
CARS: (CARS-34).
Climate-Smart Grassland Ecosystem Management Project: 10006-P166853-2021-PIR-WB-China.

### Competing Interests

The authors declare there are no competing interests.

### Author Contributions

- Xiaonan Wang conceived and designed the experiments, performed the experiments, analyzed the data, prepared figures and/or tables, authored or reviewed drafts of the article, and approved the final draft.
- Chengyang Zhou performed the experiments, prepared figures and/or tables, and approved the final draft.
- Shining Zuo performed the experiments, prepared figures and/or tables, and approved the final draft.
- Yixin Ji performed the experiments, prepared figures and/or tables, and approved the final draft.
- Wenxin Liu performed the experiments, prepared figures and/or tables, and approved the final draft.
- Ding Huang conceived and designed the experiments, authored or reviewed drafts of the article, and approved the final draft.

## DNA Deposition

The following information was supplied regarding the deposition of DNA sequences:

The 16S rRNA and ITS rRNA sequences are available at NCBI SRA: PRJNA1014180 and PRJNA1014553.

## Data Availability

The data is available at figshare: Wang, Xiaonan (2024). Soil microbial and other properties in a desert steppe of Inner Mongolia, China. figshare. Dataset. https://doi.org/10.6084/m9.figshare.25250284.v1.

## Supplemental Information

Supplemental information for this article can be found online at http://dx.doi.org/10.7717/peerj.17031#supplemental-information.

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
