# Peer review of "Heavy grazing reduces soil bacterial diversity by increasing soil pH in a semi-arid steppe"

_PeerJ, doi:10.7717/peerj.17031_

## Round 0.1 · original submission · Major Revisions

As you can see, the reviewers have raised a number of concerns. For example, the methods for data sequencing have not been well documented. Another comment on drawing conclusions from an experiment where very small amounts of dung and urine was used on a large area are of particular concern.

Reviewer 1 ·

Basic reporting

no comment

Experimental design

Experimental design lacks innovation. The determined indexes are few, especially the microbiological indexes are not enough to support the conclusion. The description of the ”Results” is not comprehensive.

Validity of the findings

Heavy grazing reduces soil bacterial diversity by increasing soil pH in a semi-arid steppe was studied based on a field grazing experiment in this study. Authors draws a conclusion of soil pH value may be the main factor driving soil microbial diversity in grazing ecosystems. Authors has measured some indexes, but not enough to support conclusions, some results have certain scientific values.

Additional comments

Heavy grazing reduces soil bacterial diversity by increasing soil pH in a semi-arid steppe was studied based on a field grazing experiment in this study. Authors draws a conclusion of soil pH value may be the main factor driving soil microbial diversity in grazing ecosystems. Authors has measured few index, some conclusions probably have certain scientific values. However, there are main problems (1) the paper lacks innovation. (2) the determined indexes are few, especially the microbiological indexes is not enough to support the conclusion. (3) The description of the “Results” is not comprehensive. The detailed review comments are as follows:

Abstract
1. Line 29-30 could be revised to “Spearman correlation analysis showed that soil pH….

Introduction
The description in the second paragraph of this paper indicted that a large number of similar studies have been carried out on the grasslands of Inner Mongolia, please supply the differences between previous finding and this paper, which can improve innovation of paper.

Materials and Methods
1. Authors obtained the rhizosphere soil by the shaking method, please describe in detail the sampling process of this method.
2. How long the soil sample was frozen before the DNA of soil microbial was extracted? Please add in the text.

Results
1. Where are the descriptions of “Effects of grazing treatment, soil position and their interaction on soil microbial diversity and chemical properties (Table 1)” and “Interaction effect between grazing intensity and soil position treatments on diversity of both bacteria and fungi in a desert steppe ecosystem in China (Fig 1)”? why did author not write these two parts at all in Result?
2. why authors did not detect bacterial and fungi community structure and composition in this study? as well as the dominant phyla? These indictors can better reflect the response of soil microorganisms to treatments.

Annotated reviews are not available for download in order to protect the identity of reviewers who chose to remain anonymous.

Reviewer 2 ·

Basic reporting

This manuscript should be improved through rearranging the details in the section of introduction and moving some contents to the section of discussion. There are too many redundant statements and citations in the introduction.
Please provide your NCBI Sequence Read Archive number in the section of Materials & Methods, which is necessary for all potential reader and reseacher.

Experimental design

The experimental design is no problem, however, simple spearman correlation analysis may not be effective to reflect their inherent relations. Please select better statistical methods, such as RDA and SEM.

I recommend supplementary determination of some organic carbon component indexes, such as labile organic carbon and mbc, relatively small amounts of dung and urine in huge area of grazing land during the short-term experimental period is unconvincing from the current results.

Validity of the findings

No more novelty findings are discovered in this manuscript.

Reviewer 3 ·

Basic reporting

no comment; The article excellently meets all the standards set.

Experimental design

Methods for the 16S and ITS data sequencing analysis has not been documented. The tools used, the filtering methods used; or the tool to calculate the diversity indices. The major part of the microbiome analysis involves alpha diversity measurement. Without known how it was performed, it is difficult to gauge how accurate they are.

Validity of the findings

no comment

Additional comments

Knowing individual taxa; and their abundances might give even better outlook. Pairing alpha diversity with beta diversities; and looking at some significantly different taxa will provide more strength to the differences in diversity conclusion.

---

## Round 0.2 · Major Revisions

I have some reservations that the comments from the previous round of revision were not fully considered by the authors. Moreover, the scope of the study doesn't look as general as suggested by the title.

Reviewer 2 ·

Basic reporting

Please the authors had better change the manuscript, because this research is only focused a specific plant species, Stipa krylovii. We do not know whether all of desert steppe plant species have similar changes. In addition, grazing activities contain feeding, nutrients recycling and trampling, which had comprehensive effects on soil properties. I suggest the authors clarify the research results based feeding and trampling.

Experimental design

There is no problem involved experimental design. Long-term field experiment is not easy for restoration ecology research.

Validity of the findings

Please the authors add some reasonable explanation in the ms before acceptance.

Additional comments

No comments.

Reviewer 3 ·

Basic reporting

Context: In the introduction, an additional line indicating why should I care about increasing or decreasing microbial diversity in the pastures would greatly increase the readability of the paper. Grazing affects microbial diversity, it affects ecosystem but why does that matter?

Experimental design

-In the rebuttal, the authors referenced another paper, though regrettably, this citation is absent within the manuscript itself. Additionally, the methods section, particularly regarding microbial diversity analysis, could be succinctly presented to provide readers with a clearer understanding. As microbial diversity is integral to the study's main results, a more transparent description of the methods would empower readers to comprehend the microbial diversity aspects more seamlessly.

- The referenced paper also uses multiple algorithms/workflows in a staggered approach. Mothur, qiime (qiime is no longer supported or was it qiime2) all have been used; and each of these workflows can perform full analysis. It shouldn't affect the paper but does raises concerns about replicability; a more streamlined process may facilitate easier replication.

Validity of the findings

No comment

---

## Round 0.3 · Minor Revisions

The revised version is technically sound and has all the changes suggested by reviewers incorporated. However, there are some minor issues of language that need to be resolved before accepting it for publication. For instance in Line 32 'the accumulation of dung and urine in livestock' needs to be revised appropriately. Moreover, the use of articles 'a/an' and 'the' should be checked carefully.

---

## Round 0.4 · accepted · Accept

Accept my felicitations for substantially improving the manuscript.